# Prospective cohort study comparing a triceps-sparing and triceps-detaching approach in total elbow arthroplasty: a protocol

Danielle Meijering ![ORCID],[1] Alexander L Boerboom,[1] Carina L E Gerritsma,[2] Bertram The,[3] Michel P J van den Bekerom,[4,5] Marco van der Pluijm,[6] Riemer J K Vegter,[7] Sjoerd K Bulstra,[1] Denise Eygendaal,[3,8] Martin Stevens,[1] On behalf of Study group

For numbered affiliations see end of article.

**Correspondence to**
Danielle Meijering;
d.meijering@umcg.nl

## ABSTRACT

**Background** New surgical approaches have been developed to optimise elbow function after total elbow arthroplasty (TEA). Currently, there is no consensus on the best surgical approach. This study aims to investigate the functional outcomes, prosthetic component position and complication rates after a triceps-sparing and a triceps-detaching approach in TEA.

**Methods and analysis** A multicentre prospective comparative cohort study will be conducted. All patients with an indication for primary TEA will enrol in either the triceps-sparing or the triceps-detaching cohort. Primary outcome measure is elbow function, specified as fixed flexion deformity. Secondary outcome parameters are self-reported and objectively measured physical functioning, including triceps force, prosthetic component position in standard radiographs and complications.

**Discussion** The successful completion of this study will clarify which surgical approach yields better functional outcomes, better prosthetic component position and lower complication rates in patients with a TEA.

**Ethics and dissemination** The Medical Ethics Review Board of University Medical Center Groningen reviewed the study and concluded that it is not clinical research with human subjects as meant in the Medical Research Involving Human Subjects Act (WMO), therefore WMO approval is not needed (METc2019/544).

**Trial registration number** NTR NL8488.

## BACKGROUND

New surgical approaches have been developed in recent decades to optimise elbow function and reduce complication rates following total elbow arthroplasty (TEA). Several surgical options exist, including a triceps-sparing approach and a triceps-detaching approach. To date, there are no prospective studies comparing triceps-sparing and triceps-detaching approaches, and full insight into the benefits and drawbacks of the two approaches is lacking.

### Strengths and limitations of this study

► This is the first prospective cohort study to compare functional outcomes, prosthetic component position and complication rates following triceps-sparing and triceps-detaching approaches in total elbow arthroplasty (TEA).

► The hospitals participating in this study represent the vast majority (over ±70%) of the TEAs annually performed in the Netherlands.

► A limitation of the study is that it is non-blinded and non-randomised.

Functional outcomes following TEA can severely deteriorate due to postoperative complications such as triceps insufficiency or long-term complications like aseptic loosening of the prosthesis.[1 2] Lenoir *et al*[3] showed that functional outcomes are also affected by prosthetic component positioning. A triceps-sparing approach has recently been advocated because of triceps insufficiency as a complication following triceps-detaching approaches. Another drawback of a triceps-detaching approach is the need to immobilise in a cast, which might impede postoperative elbow function. A triceps-sparing approach makes direct functional treatment possible, with potentially better functional outcomes.

Several studies already tried to shed light into which surgical approach yields better functional outcomes. In 2015, Dachs *et al*[4] conducted a retrospective analysis to compare triceps-sparing and triceps-detaching approaches. They concluded that elbow function, more specifically fixed flexion deformity (FFD), was better in patients following a triceps-sparing approach. They also concluded that triceps-related complications were absent in triceps-sparing approaches.

Other authors[5–7] presented similar studies with comparable results, favouring a triceps-sparing approach.

These authors, however, did not analyse prosthetic component position on radiographs. Considering the fact that the triceps-sparing approach results in less exposure of the articular surface,[8] prosthetic component position might be compromised. King et al[9] already showed that a triceps-sparing approach might lead to a more flexed position of the ulnar component. It is also known that implant malalignment increases loading patterns,[10] which might cause polyethylene wear and early loosening of the prosthesis.[11]

Hence, there is no consensus on the best surgical approach in TEA. Based on retrospective studies it can be hypothesised that a triceps-sparing approach gives favourable results in terms of functional outcomes and a lower risk of triceps insufficiency. Due to the limited visibility, prosthetic component position may be compromised though.

The aim of this study is therefore to investigate the functional outcomes, prosthetic component position in standard radiographs and complication rates following a triceps-sparing and a triceps-detaching approach in TEA.

## METHODS
### Study design
A multicentre prospective cohort study will be conducted at University Medical Center Groningen, Martini Hospital Groningen, Amphia Hospital Breda, OLVG Amsterdam and Sint Maartenskliniek Nijmegen. The hospitals participating in this study represent the vast majority (over ±70%) of the TEAs annually performed in the Netherlands. In total, 102 patients will be included. The first cohort of 51 patients will be assigned to the triceps-sparing group, the second cohort of 51 patients to the triceps-detaching group. This strategy will provide continuity in the surgical approach for surgeons and avoid surgeon-based inclusion bias.

### Recruitment and consent
All adult patients with an indication for primary TEA will be asked to participate in the study. The treating surgeon or a member of the study staff will introduce and explain the study to the patient and answer any questions the patient might have. Patients will receive written information document, and after giving informed consent they will be added to the study database.

### Study population
Inclusion criteria: age ≥18 years, primary total elbow prosthesis and ability to participate during the entire follow-up schedule. Exclusion criteria: active infection, total elbow prosthesis surgery in the past (at the ipsilateral side), that is, revision surgery, previous elbow surgery that influences function of the triceps muscle, other upper extremity injuries to the ipsilateral limb that would compromise postoperative rehabilitation, inability to follow postoperative rehabilitation (due to head injury, dementia, mental illness, etc), insufficient command of the Dutch language and conversion of surgical technique during the operation.

### Intervention
Patients will be placed in the lateral decubitus position. Tranexamic acid 1 g and cefazolin 2 g will be administered. A dorsal skin incision is made. The incision is curved to pass lateral to the olecranon tip. Full-thickness subcutaneous flaps are developed. The ulnar nerve is located and released. Then depending on the cohort, the specific surgical technique will be performed and prosthetic components will be placed. A tourniquet is inflated only during cementing. All patients will receive a linked type Latitude TEA, without radial head replacement. The radial head will not be resected unless severely damaged.

Once the tourniquet is deflated and range of motion (ROM) is assessed, the ulnar nerve is returned to its preoperative position or transposed anteriorly in case of tension to the nerve. Refixation of collateral ligaments will be performed whenever possible, and the wound is closed with sutures or staples.

Cefazolin 1 g will be given 8 and 16 hours after initial dose, tranexamic acid 1 g 8 hours after initial dose.

### Triceps sparing: cohort 1
After full-thickness skin flaps are developed, medial and lateral windows along the edge of the triceps are created. The triceps attachment remains intact to the olecranon, and after release of the collateral ligaments and opening of the joint capsule the joint can be dislocated.[12]

Active flexion, extension, pronation and supination are commenced on the first postoperative day. After 3 weeks, patients are allowed to lift 1 kg repetitively and 5 kg occasionally.

### Triceps detaching: cohort 2
A triceps-detaching technique as described by Vangorder[13] will be used. This approach reflects the aponeurosis of the musculus triceps downwards, with the base on the olecranon. The underlying muscle is longitudinally split in the midline and elevated.

Postoperatively, the elbow will be protected by a removable cast in 30° flexion for 4 weeks, avoiding active extension. This time span was chosen arbitrarily to take the postoperative healing of the triceps into account. Unfortunately, evidence for the best aftertreatment following a triceps-detaching approach is unavailable and therefore this protocol is historically rooted in our daily practice. Exercises with active flexion and passive extension are allowed, three times a day. Thereafter, the elbow will be mobilised without a brace and active triceps training is allowed. After 3 weeks, patients are allowed to lift 1 kg repetitively and 5 kg occasionally.

## Outcome measures

### Demographics
Age, sex, hand dominance, indication for surgery and previous surgery will be registered.

### Primary outcome
The primary outcome measure is elbow function, described as FFD, measured in degrees of flexion using a goniometer.

### Secondary outcomes
Secondary outcomes are self-reported physical functioning, objectively measured physical functioning, prosthetic component position on anteroposterior (AP) and lateral radiographs and complications.

### Self-reported physical functioning
► Elbow function will be measured with the Oxford Elbow Score (OES).[14] The OES consists of three domains: pain, function and social-psychological. Each domain comprises four questions with five response options per question. Each response is scored 0–4, with 0 representing greater severity. Scores for each domain are calculated as the sum of each individual item scored within that domain. These scores are then converted to a metric score between 0 and 48, where a lower score represents greater severity. The Dutch language version is considered reliable and valid.[15]

► Upper limb function will be assessed using the Quick Disabilities of the Arm, Shoulder and Hand (Quick-DASH).[16] The DASH gives a score out of 100, where a higher score indicates greater disability. The questionnaire is available in Dutch and is considered reliable and valid.[16]

► Health-related quality of life will be measured by the 5-Level version of EuroQol-5 Dimension (EQ-5D-5L),[17] a widely used and valid generic instrument to measure health-related quality of life that is validated in the Dutch language.[18 19] EQ-5D-5L has five dimensions: mobility, self-care, usual activities, pain/discomfort and anxiety/depression.Each dimension is divided into five degrees of severity: no problems, slight problems, moderate problems, severe problems and extreme problems or unable to do. Current quality of life must also be identified on the EQ-5D-5L visual analogue scale.

► Elbow pain: Level will be determined using a 10-point numeric rating scale. Pain level will be scored during activities and rest.

► Satisfaction: Patients are asked whether they are satisfied with their elbow procedure and whether they would recommend it to others. These items are self-constructed and consist of five answer options (Completely agree, Agree, Neutral, Disagree, Completely disagree). In addition, patients in the triceps-detaching cohort are asked about their satisfaction with the removable cast.

### Objectively measured physical functioning
► Active and passive ROM (flexion, extension, pronation, supination) will be measured using a goniometer. A systematic review analysing use of a goniometer in elbow measurements showed high intrarater and inter-rater reliability of the universal goniometer.[20]

► Triceps brachii force will be measured with a MicroFET and expressed in newtons. Measurements will be with the elbow in 30° flexion as described by Prkić et al.[21] In case of a flexion contracture >30°, measurement will be done in 60° flexion. In case of an acute fracture, preoperative force will be measured contralaterally. The results of the study by Prkić et al[21] will then be used to calculate the triceps brachii force on the operated elbow.

► Triceps brachii function will be measured using the Medical Research Council scale.

► Elbow stability will be stated as intact, <10° instability or >10° instability.

► Neurovascular function; motoric and sensory deficits of the ulnar, medial and radial nerves will be tested. Neurovascular function will be stated as 'intact', 'sensory deficit', 'motoric deficit', 'both sensory and motoric deficit'.

► Paraesthesia of the ulnar nerve will be measured using the Tinel test.

► Carrying angle; expressed in varus/valgus degrees and measured using a goniometer.

► Swelling of the elbow; stated as none, minimal, some, excessive.

► Pain during palpation will be tested by locating the position of the pain.

### Prosthetic component positioning
Positioning of the prosthesis will be analysed on AP and lateral radiographs at 6 weeks and 1 year of follow-up. Flexion, extension, varus and valgus positioning of both the humeral and ulnar components will be measured in degrees, as described by Lenoir et al.[3]

### Complications
Number of complications and type of complication will be registered.

## Study procedures
Clinical assessment will be performed at baseline, 6 months and 1 year after TEA (figure 1). At each follow-up visit the surgeon will conduct a physical examination and capture complications, as described previously. Patients will fill out the questionnaires at baseline and at 6 months and 1 year of follow-up. At 6 weeks and 1 year of follow-up, AP and lateral radiographs of the elbow will be taken to analyse the component position. Results of questionnaires and radiological analysis will be stored digitally in the Research Electronic Data Capture (REDCap) system. Physical examination and complications will be documented in medical records, then transported to REDCap. Data analysis will be done by an independent researcher.

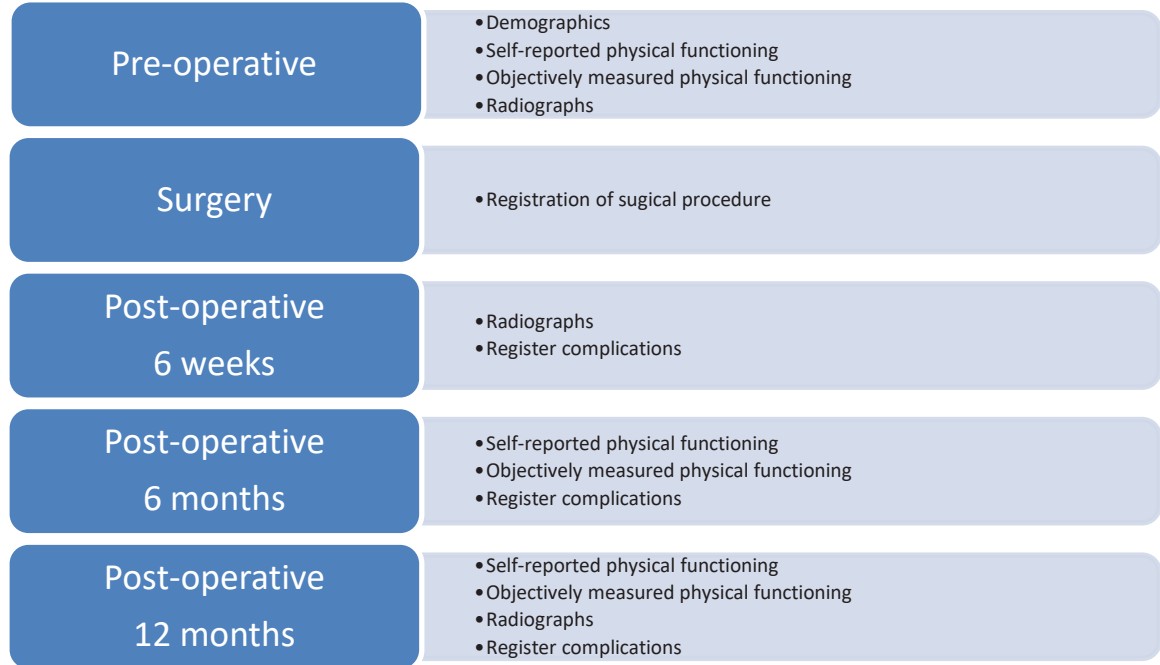

Figure 1. Study procedures

**Figure 1** Study procedures.

## Sample size calculation

To calculate the sample size, FFD is used as primary outcome measure. To detect 12° difference in elbow function, described as FFD, at 1 year of follow-up, a total of 102 patients are needed in this study. This power is based on earlier research by Dachs *et al*[4] that describes FFD after triceps-sparing and triceps-detaching surgical approaches in TEA. In this study, the difference in FFD was 12° (SD 19.6), which was considered clinically relevant, since a functional arc of motion in daily life is described as flexion-extension=120-30-0.[22] Based on this, a two-sided test with $\alpha=0.05$ and a power of 80%, a sample size of 84 patients is needed. Taking into account a 20% loss of subjects, a total of 102 patients (51 in each group) are needed.

## Statistical analysis

Descriptive statistics are used to describe patients' characteristics, clinical outcomes and scores on the questionnaires. Means and SDs will be used for continuous variables, or percentages for categorical variables. Primary outcome of the study is FFD at 1 year of follow-up. Three measurements will be collected: preoperatively and at 6 months and 1 year of follow-up. The dependency of FFD between the triceps-sparing group and the triceps-detaching group at 1 year of follow-up is our focal interest. To take into account the dependencies (the nesting structure of the three measurements nested within the patients), a longitudinal multilevel model will be used to analyse the data. This model has a level 1 (measurement) and a level 2 (patient). Relevant covariates such as age, gender and indication for surgery will be controlled for. Under the assumption of missing data being missing at random, the missing data will be imputed. The results will be considered statistically significant if $p<0.05$. SPSS statistical software (V.24.0, IBM SPSS) will be used.

## Ethics and dissemination

The Medical Ethics Review Board of University Medical Center Groningen reviewed the study and concluded that it is not clinical research with human subjects as meant in the Medical Research Involving Human Subjects Act (WMO); therefore, WMO approval is not needed (METc2019/544). Eligible patients will be informed about the study and will sign an informed consent form in order to participate. All informed consent forms will be stored in a locked research office and no personal data will be stored digitally in the REDCap system or revealed in any publication or scientific journal.

Considering current evidence, no clear preference exists for either one of the treatment protocols in this study. Both protocols are regularly applied and all surgeons participating in this study are familiar with the two surgical approaches. Management will not differ between patients, except for the surgical approach. Patients will be exposed to radiation from radiographs, but this is part of routine clinical care. No additional radiographs will be taken as part of this study. Patients may be inconvenienced by filling out questionnaires, which takes approximately 15–20 min at three different time points, but in most hospitals this too is part of routine clinical care.

## Patient and public involvement

Patients and the public were not directly involved in the development of the research question or in the design of the study.

## DISCUSSION

To date, there are no prospective studies comparing triceps-sparing and triceps-detaching approaches in TEA. Based on retrospective studies it is hypothesised that triceps-sparing approaches may lead to better elbow function and a lower risk of triceps insufficiency. However, these studies did not analyse prosthetic component position. Considering the technical nature of the triceps-sparing approach, it may carry a higher risk of component malposition, which can cause excessive loads and could lead to polyethylene wear and early loosening of the prosthesis.[11] Both triceps-sparing and triceps-detaching approaches are currently being used,[2 23] and persistently high complication rates following TEA are reported.[1 2] This stresses the need for a prospective comparative study on the best surgical approach, in order to optimise elbow function and reduce complication rates.

The decision to perform a prospective cohort study, instead of a randomised controlled trial (RCT), has been made due to two main reasons. First, we aim to include all Dutch patients requiring a total elbow prosthesis, since numbers of TEAs performed in the Netherlands are low. However, our past experience is that a substantial part of the patients is not willing to participate in an RCT. For that, we decided to run a cohort study in which in fact usual care is evaluated. By using this design, our experience is that more patients are willing to participate, which is important because the number of TEAs is already quite low. Second, a frequent switch of surgical technique is not desirable, as would be the case in an RCT. Therefore, our study design will provide continuity in the surgical approach for surgeons and for the paramedical care providers (ie, physiotherapist, nurse) during a certain period.

In recent decades, studies have reported an increasing number of TEAs performed globally.[24] This is partly due to the changing trend in indications for TEA, from primary and rheumatoid arthritis to acute trauma and post-traumatic deformities[24] and partly due to an ageing population. As the incidence of falls and fall-related injury increases with age, a further rise in the numbers of TEAs is expected.

In conclusion, currently there is no consensus on the best surgical approach in TEA and full insight into the benefits and drawbacks of the two approaches is lacking. The successful completion of this study will shed light into which surgical approach, triceps sparing or a triceps detaching, results in better functional outcomes, better prosthetic component position and lower complication rates.

Patient enrolment started in March 2020 and we expect to enrol 50 patients per year. Considering a 1 year of follow-up, publication of data are expected in 2023.

**Author affiliations**
[1]Department of Orthopedic Surgery, University Medical Center Groningen, Groningen, The Netherlands
[2]Department of Orthopaedic Surgery, Martini Hospital, Groningen, The Netherlands
[3]Department of Orthopedic Surgery, Amphia Hospital, Breda, The Netherlands
[4]Department of Orthopedic Surgery, OLVG, Amsterdam, The Netherlands
[5]Department of Human Movement Sciences, Faculty of Behavioral and Movement Sciences, Vrije Universiteit Amsterdam, Amsterdam, The Netherlands
[6]Department of Orthopedic Surgery, Sint Maartenskliniek, Nijmegen, The Netherlands
[7]Department of Human Movement Sciences, University of Groningen, Groningen, The Netherlands
[8]Department of Orthopedic Surgery, Amsterdam Universitair Medische Centra, Amsterdam, The Netherlands

**Acknowledgements** The authors thank Dr R E Stewart of UMCG for his assistance with the statistical analyses.

**Collaborators** Study group: MPJ van den Bekerom, AL Boerboom, SK Bulstra, D Eygendaal, CLE Gerritsma, P Heesterbeek, K Koenraadt, D Meijering, I van Oost, LIF Penning, M van der Pluijm, M Stevens, S Susan, B The, RJK Vegter, S Vorrink, A de Vries.

**Contributors** The study was coordinated by DM, ALB, DE and MS. MS will act as the principal investigator. DM, ALB, DE, SKB and MS developed the trial and drafted the manuscript. CLEG, BT, MPJB, MP and RJKV revised the manuscript for important intellectual content. ALB, CLEG, BT, MPJB, MP, LIF Penning and DE participated in patient inclusion. All authors have read and approved the manuscript.

**Funding** The authors have not declared a specific grant for this research from any funding agency in the public, commercial or not-for-profit sectors.

**Competing interests** DE disclosed the following financial relationships with commercial entities that produce healthcare-related products or services: institutional research support, Matthys; institutional research support, Zimmer-Biomet; institutional research support, Stryker; speaker/teacher for AO and IBRA courses; advisory board member for Lima Corporates. None related to this manuscript.

**Patient and public involvement** Patients and/or the public were not involved in the design, or conduct, or reporting, or dissemination plans of this research.

**Patient consent for publication** Not required.

**Provenance and peer review** Not commissioned; externally peer reviewed.

**ORCID iD**
Danielle Meijering http://orcid.org/0000-0001-7031-8936

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
