## [Reviewer comments · BMJ Open]

ARTICLE DETAILS

TITLE (PROVISIONAL)	A prospective cohort study comparing a triceps-sparing and triceps-detaching approach in total elbow arthroplasty: a protocol.
AUTHORS	Meijering, Danielle; Boerboom, Alexander L; Gerritsma-Bleeker, Carina; The, Bertram; van den Bekerom, Michel; van der Pluijm, Marco; Vegter, Riemer JK; Bulstra, Sjoerd; Eygendaal, Denise; Stevens, Martin

VERSION 1 – REVIEW

REVIEWER	Karunaratne, Sascha Royal Prince Alfred Hospital, Surgical Outcomes Research Centre (SOuRCe)
REVIEW RETURNED	28-Dec-2020

GENERAL COMMENTS	This manuscript presents a protocol for a prospective cohort study investigating the optimal surgical approach to optimising elbow function after primary total elbow arthroplasty (TEA), where the authors seek to compare a triceps-sparing and triceps-detaching approach. The authors' primary focus will be elbow function (fixed flexion deformity [FFD]), with secondary interest in patient-reported and objective measures of functional outcomes, prosthetic implantation positioning and complications rates. The authors provide a satisfactory argument for further investigation of these procedures given the paucity of evidence. The authors state that this study was considered to not require Wet medisch-wetenschappelijk onderzoek met mensen (WMO) approval for medical research involving human subjects (METc2019/544) and has been registered with the Dutch Trial Register (NTR NL8488). Overall, the methods presented appear sound for a prospective cohort study. The surgeons will all be allocated to group 1 (triceps-sparing) until 51 patients are recruited and then group 2 (triceps-detaching) until 51 patients are recruited (102 patients in total). In this way, surgeon-based inclusion bias will be diminished. While this cross-over of the surgery performed certainly has its merits, perhaps justification as to why a randomised controlled trial was not pursued would be pertinent. The authors present a broad inclusion criteria, followed by acceptable exclusion criteria. The surgical approaches appear to be standardised and adequately described with reference to source material that further describes the procedure. The outcomes of interest appear to be acceptable and addressing the primary and secondary goals of the study. However, the authors may consider addressing the following:
---

	 • Clarification of some of the objectively measured clinical data would be helpful (e.g. neurovascular function [Line 212], swelling of the elbow [Line 216], etc.). • Difficulty following what is being assessed or followed up and when. To resolve this, perhaps the authors can provide a figure to clarify what components are being followed up at each time point. • The authors state descriptive statistics will be used to describe patient characteristics and scores on the questionnaires (Line 250-251). The evaluation of differences and other statistics in demographics and outcome scores from baseline, to 6 months and to 1 year follow-up would be expected. It is assumed that the authors will use different tests for categorical and continuous data. Clarification should be provided for this protocol. While this reviewer believes this to be an interesting project, the authors may consider that effectiveness of procedures are best determined via randomised controlled trials. This is not to prevent the publication of this protocol, as this paper will likely contribute to our understanding. Rather, clarification as to why the authors chose not to pursue a methodology that would remove many of the biases that are inherent to cohort studies is required (accepting that enrollment has already commenced). For this reason, this reviewer would recommend the editor to accept pending addressing the above points.
--	--

REVIEWER	Jordan, Robert University Hospitals Coventry and Warwickshire NHS Trust
REVIEW RETURNED	19-Jan-2021

GENERAL COMMENTS	Congratulation on addressing an interesting topic in elbow arthroplasty and I look forward to reading the results in due course. My only comment is on the length of time spent in splint at 30 degrees flexion post-operatively (4 weeks) after triceps detachment. I appreciate the technique, length of time in splint and post-op rehab is variable between surgeons but ideally the rationale for this length of time/splinting could be included.
---

VERSION 1 – AUTHOR RESPONSE

Reviewer 1: **Comment**

Overall, the methods presented appear sound for a prospective cohort study. The surgeons will all be allocated to group 1 (tricep-sparing) until 51 patients are recruited and then group 2 (tricep-detaching) until 51 patients are recruited (102 patients in total). In this way, surgeon-based inclusion bias will be diminished. While this cross-over of the surgery performed certainly has its merits, perhaps justification as to why a randomised controlled trial was not pursued would be pertinent.

Response

The decision to perform a prospective cohort study, instead of a randomized controlled trial (RCT), has been made due to two main reasons. First, we aim to include all Dutch patients requiring a total elbow prosthesis, since numbers of TEAs performed in the Netherlands are low. However our past experience is that a substantial part of the patients is not willing to participate in a RCT. For that we decided to run a cohort study in which in fact usual care is evaluated. By using this design our experience is that more patients are willing to participate. Which is important because the number of TEAs is already quite low. Second, a frequent switch of surgical technique is not desirable, as would be the case in a RCT. Therefore, our study design will provide continuity in the surgical approach for

surgeons and for the paramedical care providers (i.e. physiotherapist, nurse) during a certain period. This has been added in line 303-313.

Comment

The authors present a broad inclusion criteria, followed by acceptable exclusion criteria. The surgical approaches appear to be standardised and adequately described with reference to source material that further describes the procedure. The outcomes of interest appear to be acceptable and addressing the primary and secondary goals of the study. However, the authors may consider addressing the following: Clarification of some of the objectively measured clinical data would be helpful (e.g. neurovascular function [Line 212], swelling of the elbow [Line 216], etc.).

Response

Thank you, we clarified as follows:

-Neurovascular function; motoric and sensory deficits of the ulnar, medial and radial nerves will be tested. Neurovascular function will be stated as “intact”, “sensory deficit”, “motoric deficit”, “both sensory and motoric deficit”.

-In our opinion we already stated the way we score “swelling of the elbow” in line 220.

Comment

Difficulty following what is being assessed or followed up and when. To resolve this, perhaps the authors can provide a figure to clarify what components are being followed up at each time point.

Response

We added a figure to clarify the study procedures (fig 1).

Comment

The authors state descriptive statistics will be used to describe patient characteristics and scores on the questionnaires (Line 250-251). The evaluation of differences and other statistics in demographics and outcome scores from baseline, to 6 months and to 1 year follow-up would be expected. It is assumed that the authors will use different tests for categorical and continuous data. Clarification should be provided for this protocol.

Response

Thank you, we clarified as follows:

Descriptive statistics are used to describe patients’ characteristics, clinical outcomes and scores on the questionnaires. Means and standard deviations (SDs) will be used for continuous variables, or percentages for categorical variables.

Comment

While this reviewer believes this to be an interesting project, the authors may consider that effectiveness of procedures are best determined via randomised controlled trials. This is not to prevent the publication of this protocol, as this paper will likely contribute to our understanding. Rather, clarification as to why the authors chose not to pursue a methodology that would remove many of the biases that are inherent to cohort studies is required (accepting that enrollment has already commenced). For this reason, this reviewer would recommend the editor to accept pending addressing the above points.

Response

Thank you, we clarified the methodology as requested previously in line 303-313.

Reviewer 2:

Comment

Congratulation on addressing an interesting topic in elbow arthroplasty and I look forward to reading the results in due course.

Response

Thank you.

Comment

My only comment is on the length of time spent in splint at 30 degrees flexion post-operatively (4 weeks) after triceps detachment. I appreciate the technique, length of time in splint and post-op rehab is variable between surgeons but ideally the rationale for this length of time/splinting could be included.

Response

We clarified as follows in line 156:

This time span was chosen arbitrary, to take the post-operative healing of the triceps into account. Unfortunately, evidence for the best after-treatment following a triceps-detaching approach is unavailable and therefore this protocol is historically rooted in our daily practice.

VERSION 2 – REVIEW

REVIEWER	Karunaratne, Sascha Royal Prince Alfred Hospital, Surgical Outcomes Research Centre (SOuRCe)
REVIEW RETURNED	31-Mar-2021

GENERAL COMMENTS	Thank you to the authors for addressing my suggestions. My point regarding justification as to why a RCT methodology was not approached was addressed. I believe having small sample available, overall concerns for compliance and feasibility of constant switching are fair justifications for the cohort approach. This may be a difference in language but with Line 312 of this new manuscript, perhaps the British-English word is spelled "Physiotherapist" rather than "Fysiotherapist" (easily amended in typesetting if an appropriate change)? For clarification of the objective criteria, I was more looking for how you'd specifically group and/or assess the degree of severity for neurovascular function and then what the difference between "minimal", "some" and "excessive" swelling is. However your response to neurovascular function appears to imply you're only interested in whether patients have or do not have a deficit; with swelling I surmise you'll use some sort of scale or report the objective criteria to make this assessment when you publish the paper. If this is true, then it is clear and no further clarification is required. Happy with the figure of data collection periods provided as well as clarification of the general statistical method. As previously stated, I look forward to seeing the outcome of this study!
--

REVIEWER	Jordan, Robert University Hospitals Coventry and Warwickshire NHS Trust
REVIEW RETURNED	22-Mar-2021

GENERAL COMMENTS	My concerns have been successfully addressed in this revised submission.
--

VERSION 2 – AUTHOR RESPONSE

Reviewer 1:

Comment

My point regarding justification as to why a RCT methodology was not approached was addressed. I believe having small sample available, overall concerns for compliance and feasibility of constant switching are fair justifications for the cohort approach.

Response

Thank you.

Comment

This may be a difference in language but with Line 312 of this new manuscript, perhaps the British-English word is spelled "Physiotherapist" rather than "Fysiotherapist" (easily amended in typesetting if an appropriate change)?

Response

The reviewer is right, the requested change has been made.

Comment

For clarification of the objective criteria, I was more looking for how you'd specifically group and/or assess the degree of severity for neurovascular function and then what the difference between "minimal", "some" and "excessive" swelling is. However your response to neurovascular function appears to imply you're only interested in whether patients have or do not have a deficit;

Response

That's correct.

Comment

with swelling I surmise you'll use some sort of scale or report the objective criteria to make this assessment when you publish the paper. If this is true, then it is clear and no further clarification is required.

Response

The reviewer is right, as suggested we will use a scale.

Comment

Happy with the figure of data collection periods provided as well as clarification of the general statistical method.

Response

Thank you.

Comment

As previously stated, I look forward to seeing the outcome of this study!

Response

Thank you!

Reviewer 2:

Comment

My concerns have been successfully addressed in this revised submission.

Response

Thank you.